# DUAL-BRANCH CENTER-SURROUNDING CONTRAST: RETHINKING CONTRASTIVE LEARNING FOR 3D POINT CLOUDS

## ABSTRACT

Most existing self-supervised learning (SSL) approaches for 3D point clouds are dominated by generative methods based on Masked Autoencoders (MAE). However, these generative methods have been proven to struggle to capture high-level discriminative features effectively, leading to poor performance on linear probing and other downstream tasks. In contrast, contrastive methods excel in discriminative feature representation and generalization ability on image data. Despite this, contrastive learning (CL) in 3D data remains scarce. Besides, simply applying CL methods designed for 2D data to 3D fails to effectively learn 3D local details. To address these challenges, we propose a novel Dual-Branch **C**enter-**S**urrounding **Con**trast (CSCon) framework. Specifically, we apply masking to the center and surrounding parts separately, constructing dual-branch inputs with center-biased and surrounding-biased representations to better capture rich geometric information. Meanwhile, we introduce a patch-level contrastive loss to further enhance both high-level information and local sensitivity. Under the FULL and ALL protocols, CSCon achieves performance comparable to generative methods; under the MLP-LINEAR, MLP-3, and ONLY-NEW protocols, our method attains state-of-the-art results, even surpassing cross-modal approaches. In particular, under the MLP-LINEAR protocol, our method outperforms the baseline (Point-MAE) by **7.9%**, **6.7%**, and **10.3%** on the three variants of ScanObjectNN, respectively. The code will be made publicly available.

## 1 INTRODUCTION

Self-supervised learning (SSL), which enables the extraction of intrinsic data structures and semantics without manual annotation, has attracted increasing attention in 3D point cloud representation learning in recent years. Currently, SSL for 3D point clouds can be broadly categorized into generative approaches (Achlioptas et al., 2018; Pang et al., 2022; Zhang et al., 2022b; Zha et al., 2024; Zhang et al., 2024; Ren et al., 2024; Su et al., 2025) and contrastive approaches (Xie et al., 2020; Sanghi, 2020; Dong et al., 2022; Du et al., 2021; Huang et al., 2021; Afham et al., 2022). Generative methods, typically based on masked autoencoders, aim to capture low-level features by reconstructing the masked original input. Contrastive methods, on the other hand, improve the high-level discriminative power of global representations by pulling together features of the same instance under different augmentations and pushing apart features of different instances.

Despite their prevalence, current generative SSL methods (Pang et al., 2022; Zhang et al., 2022b; Ren et al., 2024; Zhang et al., 2025; Su et al., 2025; Lin et al., 2025; Wang et al., 2025; Cheng et al., 2025) exhibit a significant limitation. By concentrating on reconstructing masked points, they often struggle to capture semantically rich, high-level features, which can lead to suboptimal performance on downstream tasks like object classification (as evidenced by linear probing results in Table 3).

While contrastive learning inherently excels at building discriminative representations, its application in the 3D domain remains underexplored and faces critical challenges. Existing 3D contrastive methods (Xie et al., 2020; Afham et al., 2022; Qi et al., 2023) typically apply the InfoNCE loss (Oord et al., 2018; Chen et al., 2020) by treating partial scans from the same object as positive pairs. However, this global-level approach often fails to capture fine-grained local structures, resulting in a loss of critical detail and underwhelming performance in fine-tuning scenarios (see Table 2).

Table 1: Methodology comparisons of the proposed CSCon and previous advanced methods.

| Methods | Framework | #Params (M) | Local | Objective | Performance on ScanObjectNN | |
|---|---|---|---|---|---|---|
| | | | | | Avg. LINEAR | Avg. MLP-3 |
| Point-MAE (Pang et al., 2022) | Generative | 29.0 (baseline) | ✓ | low-level | 79.7±0.4 | 82.3±0.5 |
| Point-FEMAE (Zha et al., 2024) | Generative | 41.5 (1.43×) | ✓ | low-level | 86.3±0.2 | 87.1±0.6 |
| PCP-MAE (Zhang et al., 2024) | Generative | 29.5 (1.02×) | ✓ | low-level | 86.5±0.2 | 88.1±0.4 |
| Point-PQAE (Zhang et al., 2025) | Generative | 29.5 (1.02×) | ✓ | low-level | 86.7±0.3 | 88.3±0.2 |
| ReCon (Qi et al., 2023) | Generative | 140.9 (4.85×) | ✓ | low-level | 86.9±0.2 | 88.4±0.3 |
| CrossPoint (Afham et al., 2022) | Contrastive | 27.7 (0.96×) | ✗ | high-level | 81.7 | ∼ |
| CSCon (Ours) | Contrastive | **22.1 (0.76×)** | ✓ | high-level | **88.0±0.3** | **89.8±0.3** |

To simultaneously address the need for both high-level discriminative power and local structural sensitivity, we propose a novel Dual-Branch Center-Surrounding Contrastive learning framework, termed CSCon. Unlike conventional contrastive methods that rely on multi-view pre-training, CSCon operates within a single-view paradigm, eliminating the need for a decoder and significantly reducing computational overhead. The main contributions of this work can be summarized as follows:

**i) A novel contrastive learning paradigm for 3D.** Instead of simply applying contrastive objectives in 2D, which requires complex and well-designed augmentation to generate views, inspired by the property of point clouds, we propose to partition the patches into center and surrounding parts, and treat the two incomplete parts as positives, forcing the network to learn rich geometric information.

**ii) Inner-sample Patch-Level Contrastive Loss.** Unlike prior works (Qi et al., 2023; Xie et al., 2020), which directly employs a global contrastive loss (proven ineffective for capturing local information), we propose a patch-level contrastive loss, treating the surrounding part and center part from the same patches and different patches (in one single point cloud) as positives and negatives, respectively.

**iii) Advancing the State-of-the-Art (SOTA).** CSCon achieves new SOTA results on multiple benchmarks with much fewer parameters (does not require decoders). Under the MLP-LINEAR, MLP-3, and ONLY-NEW protocols, our method significantly outperforms existing approaches; for the FULL and ALL protocols, CSCon achieves competitive results with the best-performing models.

## 2 RELATED WORK

**Self-Supervised Learning (SSL) for image.** The primary objective of SSL in computer vision is to leverage large-scale unlabeled data by designing pretext tasks for pre-training, thereby enabling models to automatically learn discriminative image representations. Current mainstream SSL approaches can be broadly categorized into contrastive learning (CL) (Zhang et al., 2023b) and masked image modeling (MIM) (Bao et al., 2021; Chen et al., 2024). Contrastive learning methods aim to maximize the consistency of representations for the same data instance under different views, effectively uncovering the underlying structure within unlabeled data. Representative contrastive learning methods include SimCLR (Chen et al., 2020), which introduces data augmentation strategies, DINO (Zhang et al., 2022a), which adapts self-distillation to unsupervised learning, and Rank-N-Contrast (RNC) (Zha et al., 2023), which proposes a ranking loss based on sequential representation learning. On the other hand, MIM methods partially mask the input image and require the model to reconstruct the masked regions from the visible parts, thereby encouraging the learning of more discriminative features. MAE (He et al., 2022), as a representative method, utilizes an autoencoder architecture to reconstruct masked regions. Building upon this, (Amac et al., 2022; Xie et al., 2022) novel pre-training frameworks, while (Wei et al., 2022; Baevski et al., 2022; Dong et al., 2023) further introduce new reconstruction objectives.

**SSL for Point Clouds.** Benefiting from the remarkable success of the SSL paradigm in the 2D image domain, this paradigm has recently been extensively explored and applied to 3D point cloud data. However, the inherent sparsity and irregularity of point clouds pose significant challenges for SSL. Autoencoder-based (Achlioptas et al., 2018; Pang et al., 2022; Zhang et al., 2022b; Zha et al., 2024; Zhang et al., 2024; Ren et al., 2024; Gao et al., 2025) reconstruction methods primarily focus on recovering the global structure, but their ability to capture fine-grained local geometric details remains limited. In contrast to reconstruction approaches that aim to minimize geometric discrepancies between input and output, CL (Xie et al., 2020; Sanghi, 2020; Dong et al., 2022; Du

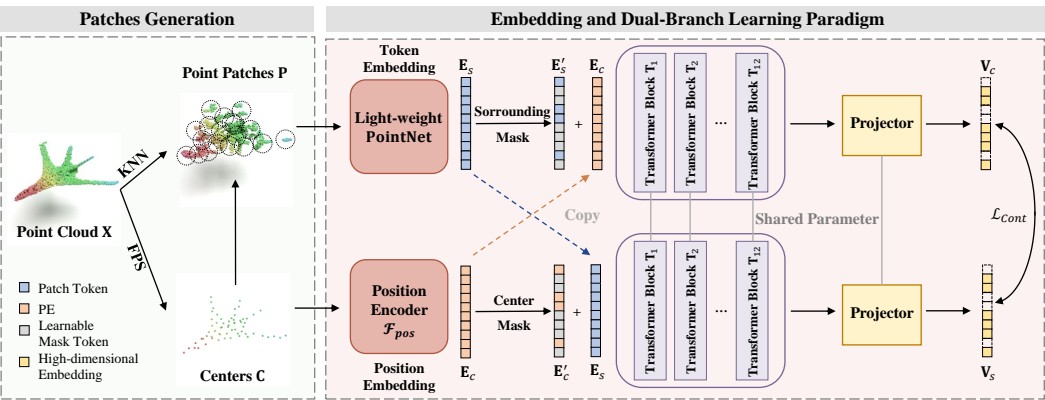

Figure 1: Illustration of the framework of the proposed CSCon. Both the transformer blocks and the projector share the parameters in two branches. Note that the point patches $\mathbf{P}$ only contain local information, and the center positions (absolute xyz coordinate of the center points) are discarded.

et al., 2021; Huang et al., 2021; Afham et al., 2022) is fundamentally driven by the principle of instance discrimination. PointContrast (Xie et al., 2020) is the first to introduce CL into the point cloud domain, aiming to learn dense point-level feature consistency across different views. Building upon this, CrossPoint (Afham et al., 2022) further incorporates cross-modal global feature contrast, which effectively enhances the global semantic comprehension of the model. ReCon (Qi et al., 2023) ingeniously integrates generative and contrastive frameworks, achieving a complementary combination of their respective strengths. PoCCA (Wu et al., 2025) enables information exchange between different branches within a single-modal framework by leveraging cross-attention fusion.

## 3 THE PROPOSED CSCON

### 3.1 PATCHES GENERATION AND EMBEDDING

**Patches Generation.** Patch-based paradigms (Pang et al., 2022; Zha et al., 2024; Zhang et al., 2024) have been widely adopted in the field of point clouds self-supervised learning. In our approach (see Fig.1), the input point cloud sample is partitioned into multiple point cloud patches using Farthest Point Sampling (FPS) (Qi et al., 2017b) and the k-Nearest Neighbors (KNN) (Cover & Hart, 1967) algorithms. Specifically, given a point cloud data $\mathbf{X} \in \mathbb{R}^{p \times 3}$, we first employ FPS to iteratively select $N$ representative center points $\mathbf{C}$, where $\mathbf{C} \in \mathbf{N} \times \mathbf{3}$. Subsequently, for each center point, we utilize KNN to identify its $k$ nearest neighbors, thereby yeilding point cloud patches $\mathbf{P} \in \mathbb{R}^{N \times k \times 3}$,

$$\mathbf{C} = FPS(\mathbf{X}), \ \mathbf{P} = KNN(\mathbf{X}, \mathbf{C}), \ \mathbf{C} \in \mathbb{R}^{N \times 3}, \ \mathbf{P} \in \mathbb{R}^{N \times k \times 3} \quad (1)$$

Similar to previous approaches (Pang et al., 2022), to achieve local coordinate consistency, we normalize the coordinates of each point within a patch with respect to its corresponding center point.

**Latent embeddings of centers and surrounding points.** The coordinates of the center point have been proven to contain rich geometric and semantic information, serving as a fundamental basis for point cloud understanding (Zhang et al., 2024). Therefore, we introduce a simple position projector (composed of a two-layer learnable MLP) $\mathcal{F}_{pos}$ that maps the center point coordinates $\mathbf{C}$ into latent space, yielding the final position embedding $\mathbf{E}_c \in \mathbb{R}^{N \times D}$, where $D$ means the output dimension:

$$\mathbf{E}_c = \mathcal{F}_{pos}(\mathbf{C}), \ \mathbf{E}_c \in \mathbb{R}^{N \times D} \quad (2)$$

Similar to Point-MAE Pang et al. (2022), we employ a lightweight PointNet (Qi et al., 2017a) to encode each patch $\mathbf{P}$, yielding the high-dimensional tokens $\mathbf{E}_s$ to represent the surrounding points:

$$\mathbf{E}_s = \text{PointNet}(\mathbf{P}), \ \mathbf{E}_s \in \mathbb{R}^{N \times D} \quad (3)$$

In line with PCP-MAE, PointNet$(\cdot)$ is randomly initialized and learned through gradient signal.

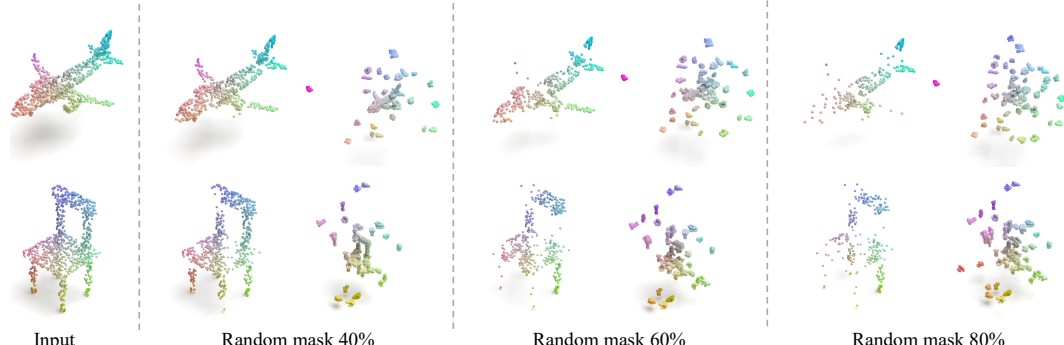

Input               Random mask 40%            Random mask 60%            Random mask 80%

Figure 2: **Visualization on the ShapeNet validation set.** The leftmost column shows the ground truth, while the subsequent columns present dual-branch inputs under different masking ratios: the left side displays the masked surrounding points, and the right side shows the masked center points. Note that when the center is masked, **the patch loses its global coordinates and retains only the local features from the normalized surrounding points**. Therefore, for patches with masked center, we assign randomly generated center coordinates composed of noise for better visualization.

## 3.2 DUAL-BRANCH LEARNING PARADIGM

After obtaining the latent representations of center points $\mathbf{E}_c = [\mathbf{e}_c^1, \mathbf{e}_c^2, \cdots, \mathbf{e}_c^N]$ and their corresponding sorrounding points $\mathbf{E}_s = [\mathbf{e}_s^1, \mathbf{e}_s^2, \cdots, \mathbf{e}_s^N]$, where $\mathbf{e}_c^i, \mathbf{e}_s^i \in \mathbb{R}^{1 \times D}$, we first initialize two learnable mask vector $\mathbf{m}_c$ and $\mathbf{m}_s$. Then, similar to the previous masking strategy (Pang et al., 2022), we randomly mask $M$ patches with a pre-defined masking ratio. For simplicity, we assume the last $M$ patches in the sequence $\mathbf{E}_c$ and $\mathbf{E}_s$ are masked. Therefore, the sequences can be written as:

$$\mathbf{E}_c' = \text{CenterMask}(\mathbf{E}_c) = \left[\mathbf{e}_c^1, \mathbf{e}_c^2, \cdots, \mathbf{e}_c^{N-M}, \mathbf{m}_c, \mathbf{m}_c, \cdots, \mathbf{m}_c\right]$$
$$\mathbf{E}_s' = \text{SorroundingMask}(\mathbf{E}_s) = \left[\mathbf{e}_s^1, \mathbf{e}_s^2, \cdots, \mathbf{e}_s^{N-M}, \mathbf{m}_s, \mathbf{m}_s, \cdots, \mathbf{m}_s\right]$$
(4)

After obtaining the masked version $\mathbf{E}_c'$ and $\mathbf{E}_s'$ of the two raw sequence $\mathbf{E}_c$ and $\mathbf{E}_s$, we calculate the input sequences by (input patches usually contains positional embeddings and patch embeddings):

$$\mathbf{Z}_c = \mathbf{E}_c' + \mathbf{E}_s = \left[\mathbf{e}_c^1 + \mathbf{e}_s^1, \mathbf{e}_c^2 + \mathbf{e}_s^2, \cdots, \mathbf{e}_c^{N-M} + \mathbf{e}_s^{N-M}, \mathbf{m}_c + \mathbf{e}_s^{N-M+1}, \cdots, \mathbf{m}_c + \mathbf{e}_s^N\right]$$
$$\mathbf{Z}_s = \mathbf{E}_s' + \mathbf{E}_c = \left[\mathbf{e}_s^1 + \mathbf{e}_c^1, \mathbf{e}_s^2 + \mathbf{e}_c^2, \cdots, \mathbf{e}_s^{N-M} + \mathbf{e}_c^{N-M}, \mathbf{m}_s + \mathbf{e}_c^{N-M+1}, \cdots, \mathbf{m}_s + \mathbf{e}_c^N\right]$$
(5)

where $\mathbf{Z}_c$ and $\mathbf{Z}_s$ mean we only partially mask the center part and partially mask the surrounding points of the input sequence, respectively. We visualize the inputs to the dual-branch network (see Figure 2). Finally, the two calculated input sequences will be fed into the encoder $f_\theta$, following a projector (commonly used in previous contrastive methods), yielding two output representations $\mathbf{H}_c = [\mathbf{h}_c^1, \mathbf{h}_c^2, \cdots, \mathbf{h}_c^N]$ and $\mathbf{H}_s = [\mathbf{h}_s^1, \mathbf{h}_s^2, \cdots, \mathbf{h}_s^N]$, respectively.

## 3.3 INNER-INSTANCE PATCH-LEVEL CONTRASTIVE LOSS

Recall that our goal is to learn high-level and discriminative 3D representations. Therefore, we aim to maximize the mutual information between the high-dimensional patch embeddings and center embeddings. Specifically, denote the center-only representations and the surrounding-only representations as $\mathbf{V}_s$ and $\mathbf{V}_c$, respectively, where the two representations are learned by the encoder. Supposed the last $M$ tokens are partially masked, and $[:, M :]$ indicates the last $M$ tokens, we have:

$$\mathbf{V}_c = \mathbf{H}_c[:, M :] = (\mathbf{h}_c^{N-M+1}, \mathbf{h}_c^{N-M+2}, ..., \mathbf{h}_c^N)$$
$$\mathbf{V}_s = \mathbf{H}_s[:, M :] = (\mathbf{h}_s^{N-M+1}, \mathbf{h}_s^{N-M+2}, ..., \mathbf{h}_s^N)$$
(6)

Intuitively, we aim to maximize the consistency between $\mathbf{V}_c$ and $\mathbf{V}_s$ if they originate from the same masked patch. Therefore, we propose an inner-instance patch-level contrastive loss:

$$\mathcal{L}_{Cont} = -\frac{1}{M} \left[ \sum_{i=1}^{M} \log \frac{e^{sim(\mathbf{v}_c^i, \mathbf{v}_s^i)/\tau}}{\sum_{j=1}^{M} e^{sim(\mathbf{v}_c^i, \mathbf{v}_s^j)/\tau}} \right]$$
(7)

Table 2: Classification Accuracy on Real-scanned (ScanObjectNN) and Synthetic (ModelNet40) Point Clouds under the **FULL** Protocol. In ScanObjectNN, we report the accuracy (%) on three variants. For ModelNet40, we present the accuracy (%) on both 1K and 8K points. #Params denotes the number of parameters in the inference model. The ddag ($\ddag$) indicates results from ReCon (Qi et al., 2023), which uses cross-modal contrastive learning (CMC) with InfoNCE loss.

| Methods | #Params (M) | ScanObjectNN | | | ModelNet40 | |
| --- | --- | --- | --- | --- | --- | --- |
| | | OBJ_BG | OBJ_ONLY | PB_T50_RS | 1K P | 8K P |
| *Supervised Learning Only* | | | | | | |
| PointNet (Qi et al., 2017a) | 3.5 | 73.3 | 79.2 | 68.0 | 89.2 | 90.8 |
| PointNet++ (Qi et al., 2017b) | 1.5 | 82.3 | 84.3 | 77.9 | 90.7 | 91.9 |
| DGCNN (Wang et al., 2019) | 1.8 | 82.8 | 86.2 | 78.1 | 92.9 | - |
| SimpleView (Goyal et al., 2021) | - | - | - | 80.5±0.3 | 93.9 | - |
| MVTN (Hamdi et al., 2021) | 11.2 | 92.6 | 92.3 | 82.8 | 93.8 | - |
| PointMLP (Ma et al., 2022) | 12.6 | - | - | 85.4±0.3 | 94.5 | - |
| P2P-HorNet (Wang & Yoon, 2021) | 195.8 | - | - | 89.3 | 94.0 | - |
| *with Self-Supervised Representation Learning (FULL)* | | | | | | |
| *Methods using only single-modal information* | | | | | | |
| PointContrast (Xie et al., 2020) | 8.6 | 76.68 | 77.43 | 66.69 | - | - |
| Point-BERT (Yu et al., 2022) | 22.1 | 87.43 | 88.12 | 83.07 | 93.2 | 93.8 |
| MaskPoint (Liu et al., 2022) | - | 89.30 | 88.10 | 84.30 | 93.8 | - |
| Point-MAE (Pang et al., 2022) | 22.1 | 90.02 | 88.29 | 85.18 | 93.8 | 94.0 |
| Point-M2AE (Zhang et al., 2022b) | 15.3 | 91.22 | 88.81 | 86.43 | 94.0 | - |
| PointGPT (Chen et al., 2023) | 19.5 | 91.60 | 90.00 | 86.90 | 94.0 | 94.2 |
| PointDif (Zheng et al., 2024) | 22.1 | 91.91 | 93.29 | 87.61 | - | - |
| Point-FEMAE (Zha et al., 2024) | 27.4 | 95.18 | 93.29 | 90.22 | **94.5** | - |
| Point-CMAE (Ren et al., 2024) | 22.1 | 93.46 | 91.05 | 88.75 | 93.6 | - |
| Point-PQAE (Zhang et al., 2025) | 22.1 | 95.00 | **93.60** | 89.6 | 94.0 | **94.3** |
| DAP-MAE (Gao et al., 2025) | - | 95.18 | 93.45 | 90.25 | - | - |
| PointLAMA (Lin et al., 2025) | 12.8 | 94.51 | 92.86 | 89.53 | 94.5 | - |
| **CSCon (Ours)** | 22.1 | **95.35** | 92.77 | **90.42** | 94.1 | **94.3** |
| *Improve (over Point-MAE)* | - | +5.13 | +4.48 | +5.24 | +0.3 | +0.3 |
| *Methods using cross-modal information and teacher models* | | | | | | |
| ACT (Dong et al., 2022) | 22.1 | 93.29 | 91.91 | 88.21 | 93.7 | 94.0 |
| Joint-MAE (Guo et al., 2023) | - | 90.94 | 88.86 | 86.07 | 94.0 | - |
| I2P-MAE (Zhang et al., 2023a) | 15.3 | 94.14 | 91.57 | 90.11 | 94.1 | - |
| TAP (Wang et al., 2023) | 22.1 | 90.36 | 89.50 | 85.67 | - | - |
| CMC$^\ddag$ (Qi et al., 2023) | - | - | - | 82.48 | - | - |
| ReCon (Qi et al., 2023) | 43.6 | 95.18 | 93.63 | 90.63 | 94.5 | 94.7 |
| UniPre3D (Wang et al., 2025) | - | 92.60 | 92.08 | 87.93 | - | - |

where $sim(\mathbf{v}_c, \mathbf{v}_s) = \mathbf{v}_c \cdot \mathbf{v}_s / (\|\mathbf{v}_c\|_2 \cdot \|\mathbf{v}_s\|_2)$ denotes the cosine similarity between vectors $\mathbf{v}_c$ and $\mathbf{v}_s$, and $\tau$ is a temperature hyper-parameter. For simplicity, we set $\tau = 1$ by default. Eq. 7 can be also decomposed into two parts, i.e., alignment and uniformity (Wang & Isola, 2020), where the alignment part aims to maximize the similarity of the positive pairs. In this paper, $\mathbf{v}_c^i$ and $\mathbf{v}_s^i$ are both learned through the $i$-th patches. Therefore, we treat them as a positive pair to learn patch-level information.

## 4 EXPERIMENTS

In this section, we first pre-train our model on ShapeNet (Chang et al., 2015). Subsequently, we transfer the pre-trained model under various evaluation protocols to adapt it to different downstream tasks, including object classification, few-shot learning, and part segmentation. Experimental results demonstrate that CSCon achieves SOTA performance on several tasks, fully showcasing its strong generalization ability and advanced capabilities. Finally, we design and perform ablation studies to further validate the effectiveness of each component in our proposed method. Detailed evaluation protocols are provided in Appendix A.

### 4.1 TRANSFER LEARNING ON DOWNSTREAM TASKS

**3D Real-World Object Classification.** To comprehensively evaluate model transferability, we conduct classification experiments on the real-world 3D object dataset ScanObjectNN (Uy et al., 2019), which is well-known for its challenging classification tasks. ScanObjectNN contains approximately

Table 3: Classification Accuracy on Real-scanned (ScanObjectNN) and Synthetic (ModelNet40) Point Clouds under the **MLP-LINEAR** and **MLP-3** Protocols. In ScanObjectNN, we report the accuracy (%) on three variants. For ModelNet40, we present the accuracy (%) on both 1K and 8K points. #Params denotes the number of parameters in the inference model. $^{\dagger}$ indicates that the result is taken from the ReCon baseline, where the rotation data augmentation strategy is applied.

| Methods | #Params (M) | ScanObjectNN | | | ModelNet40 | |
|---|---|---|---|---|---|---|
| | | OBJ_BG | OBJ_ONLY | PB_T50_RS | 1K P | 8K P |
| *with Self-Supervised Representation Learning* $(MLP-LINEAR)$ | | | | | | |
| *Methods using only single-modal information* | | | | | | |
| Point-MAE (Pang et al., 2022) | 22.1 | 82.6±0.6 | 83.5±0.4 | 73.1±0.3 | 91.2±0.3 | - |
| Point-MAE$^{\dagger}$ (Pang et al., 2022) | 22.1 | 82.8±0.3 | 83.2±0.2 | 74.1±0.2 | 90.2±0.1 | 90.7±0.1 |
| Point-FEMAE (Zha et al., 2024) | 22.1 | 89.0±0.2 | 89.5±0.2 | 80.3±0.1 | **92.4±0.1** | - |
| PCP-MAE (Zhang et al., 2024) | 22.1 | 89.4±0.1 | 89.4±0.3 | 80.6±0.1 | **92.4±0.3** | - |
| Point-CMAE (Ren et al., 2024) | 22.1 | 83.5±0.3 | 83.5±0.4 | 73.2±0.1 | 92.3±0.3 | - |
| Point-PQAE (Zhang et al., 2025) | 22.1 | 89.3±0.3 | 90.2±0.4 | 80.8±0.1 | 92.0±0.2 | 92.2±0.1 |
| **CSCon (Ours)** | 22.1 | **90.5±0.3** | **90.2±0.4** | **83.4±0.2** | **92.4±0.1** | **93.0±0.2** |
| *Improve (over Point-MAE$^{\dagger}$)* | - | +7.7 | +7.0 | +9.3 | +2.2 | +2.3 |
| *Methods using cross-modal information and teacher models* | | | | | | |
| ACT (Dong et al., 2022) | 22.1 | 85.2±0.8 | 85.8±0.2 | 76.3±0.3 | 91.4±0.2 | 91.8±0.2 |
| ReCon (Qi et al., 2023) | 43.6 | 89.5±0.2 | 89.7±0.2 | 81.4±0.1 | 92.5±0.2 | 92.7±0.1 |
| *with Self-Supervised Representation Learning* $(MLP-3)$ | | | | | | |
| *Methods using only single-modal information* | | | | | | |
| Point-MAE (Pang et al., 2022) | 22.1 | 84.3± 0.6 | 85.2 ±0.7 | 77.3 ±0.1 | 92.3 ±0.1 | - |
| Point-MAE$^{\dagger}$ (Pang et al., 2022) | 22.1 | 85.8±0.3 | 85.5±0.2 | 80.4±0.2 | 91.3±0.2 | 91.7±0.2 |
| Point-FEMAE (Zha et al., 2024) | 22.1 | 88.5±0.9 | 89.5±0.6 | 83.3±0.4 | 92.4±0.1 | - |
| PCP-MAE (Zhang et al., 2024) | 22.1 | 90.1±0.3 | **91.0±0.4** | 83.2 ±0.4 | 92.9±0.1 | - |
| Point-CMAE (Ren et al., 2024) | 22.1 | 85.9±0.5 | 85.6 ±0.4 | 77.5 ±0.1 | 92.6±0.2 | - |
| Point-PQAE (Zhang et al., 2025) | 22.1 | 90.7±0.2 | 90.9±0.2 | 83.3±0.1 | 92.8±0.1 | 92.9±0.1 |
| **CSCon (Ours)** | 22.1 | **92.3±0.4** | **91.0±0.3** | **86.2±0.2** | **93.1±0.1** | **93.7±0.2** |
| *Improve (over Point-MAE$^{\dagger}$)* | - | +6.5 | +5.5 | +5.8 | +1.8 | +2.0 |
| *Methods using cross-modal information and teacher models* | | | | | | |
| ACT (Dong et al., 2022) | 22.1 | 87.1±0.2 | 87.9±0.4 | 81.5±0.2 | 92.7±0.2 | 93.0±0.1 |
| ReCon (Qi et al., 2023) | 43.6 | 90.6±0.2 | 90.7±0.3 | 83.8±0.4 | 93.0±0.1 | 93.4±0.1 |

15,000 real objects spanning 15 categories. We transfer the pretrained model to ScanObjectNN and evaluate it under three commonly used settings: OBJ-BG, OBJ-ONLY, and PB-T50-RS. The experimental results are summarized in Table 2 and Table 3. Under the FULL protocol, CSCon achieves comparable results to the SOTA unimodal self-supervised methods (Zhang et al., 2024) across all settings, and demonstrates competitive results with leading cross-modal approaches, particularly on the OBJ-BG and PB-T50-RS settings. Compared to the most relevant baseline, Point-MAE, CSCon improves accuracy by 5.13%, 4.48%, and 5.24% under the three respective settings. Notably, under the MLP-LINEAR and MLP-3 protocols, our method also significantly outperforms all unimodal and cross-modal methods, achieving substantial gains over the baseline Point-MAE$^{\dagger}$, which further demonstrates the superiority of CSCon in transfer learning scenarios.

**3D Clean Object Classification.** We systematically evaluate the 3D object classification capability of our pre-trained model on the ModelNet40 dataset. ModelNet40 consists of 12K high-quality 3D CAD models spanning 40 distinct categories, and is widely adopted as a standard benchmark in the field of 3D object recognition. During training, we employ the Scale & Translate data augmentation strategy. As shown in Table 2 and Table 3. Under both the FULL and MLP-LINEAR protocols, CSCon achieves competitive or superior performance across all metrics. Notably, under the MLP-3 protocol, our method surpasses the Point-MAE$^{\dagger}$ baseline by 1.8% and 2.0%, respectively.

**Few-shot learning.** To evaluate the generalization capability of our method, we conduct few-shot learning experiments on ModelNet40, adhering to the standard "w-way, s-shot" protocol (Sharma & Kaul, 2020; Xie et al., 2020; Dong et al., 2022). As shown in Table 4 and Table 5, our method achieves new state-of-the-art performance across all settings, demonstrating superior generalization and feature extraction capabilities. Notably, CSCon outperforms all existing single-model and cross-modal approaches. Under the 5-way 10-shot and 10-way 20-shot settings, our method secures the top results. The advantage is particularly pronounced under the linear evaluation protocol (MLP-

Table 4: Few-shot classification results on ModelNet40 under the **FULL** Protocol. We perform 10 trials for each experimental setting and the mean accuracy (%) and standard deviation are reported.

| Methods | 5-way | | 10-way | |
|---|---|---|---|---|
| | 10-shot | 20-shot | 10-shot | 20-shot |
| *Supervised Learning Only* | | | | |
| PointNet (Qi et al., 2017a) | 52.0±3.8 | 57.8±4.9 | 46.6±4.3 | 35.2±4.8 |
| DGCNN (Wang et al., 2019) | 31.6±2.8 | 40.8±4.6 | 19.9±2.1 | 16.9±1.5 |
| OcCo (Wang et al., 2021) | 90.6±2.8 | 92.5±1.9 | 82.9±1.3 | 86.5±2.2 |
| *with Self-Supervised Representation Learning (FULL)* | | | | |
| *Methods using only single-modal information* | | | | |
| Point-BERT (Yu et al., 2022) | 94.6±3.1 | 96.3±2.7 | 91.0±5.4 | 92.7±5.1 |
| MaskPoint (Liu et al., 2022) | 95.0±3.7 | 97.2±1.7 | 91.4±4.0 | 93.4±3.5 |
| Point-MAE (Pang et al., 2022) | 96.3±2.5 | 97.8±1.8 | 92.6±4.1 | 95.0±3.0 |
| Point-M2AE (Zhang et al., 2022b) | 96.8±1.8 | 98.3±1.4 | 92.3±4.5 | 95.0±3.0 |
| PointGPT (Chen et al., 2023) | 96.8±2.0 | 98.6±1.1 | 92.6±4.6 | 95.2±3.4 |
| Point-FEMAE (Zha et al., 2024) | 97.2±1.9 | 98.6±1.3 | 94.0±3.3 | 95.8±2.8 |
| PCP-MAE (Zhang et al., 2024) | 97.4±2.3 | 99.1±0.8 | 93.5±3.7 | 95.9±2.7 |
| Point-CMAE (Ren et al., 2024) | 96.7±2.2 | 98.0±0.9 | 92.7±4.4 | 95.3±3.3 |
| Point-PQAE (Zhang et al., 2025) | 96.9±3.2 | 98.9±1.0 | **94.1±4.2** | 96.3±2.7 |
| DAP-MAE (Gao et al., 2025) | **97.5±1.8** | 98.9±0.6 | 93.3±3.9 | 95.2±2.8 |
| PointLAMA (Lin et al., 2025) | 97.2±1.9 | 99.0±0.9 | 94.0±4.5 | 95.8±3.1 |
| **CSCon (Ours)** | **97.5±2.2** | **99.4±0.8** | 93.6±3.7 | **96.5±2.2** |
| *Improve (over Point-MAE)* | *+1.2* | *+1.6* | *+1.0* | *+1.5* |
| *Methods using cross-modal information and teacher models* | | | | |
| ACT (Dong et al., 2022) | 96.8±2.3 | 98.0±1.4 | 93.3±4.0 | 95.6±2.8 |
| Joint-MAE (Guo et al., 2023) | 96.7±2.2 | 97.9±1.9 | 92.6±3.7 | 95.1±2.6 |
| I2P-MAE (Zhang et al., 2023a) | 97.0±1.8 | 98.3±1.3 | 92.6±5.0 | 95.5±3.0 |
| TAP (Wang et al., 2023) | 97.3±1.8 | 97.8±1.9 | 93.1±2.6 | 95.8±1.0 |
| ReCon (Qi et al., 2023) | 97.3±1.9 | 98.9±1.2 | 93.3±3.9 | 95.8±3.0 |

Table 5: Few-shot classification results on ModelNet40 under the **MLP-LINEAR** and **MLP-3** Protocols. We perform 10 trials for each experimental setting. Accuracy and deviation are reported.

| Methods | 5-way | | 10-way | |
|---|---|---|---|---|
| | 10-shot | 20-shot | 10-shot | 20-shot |
| *with Self-Supervised Representation Learning (MLP − LINEAR)* | | | | |
| Point-MAE (Pang et al., 2022) | 91.1±5.6 | 91.7±4.0 | 83.5±6.1 | 89.7±4.1 |
| ACT (Dong et al., 2022) | 91.8±4.7 | 93.1±4.2 | 84.5±6.4 | 90.7±4.3 |
| ReCon (Qi et al., 2023) | **96.9±2.6** | 98.2±1.4 | **93.6±4.7** | 95.4±2.6 |
| Point-FEMAE (Zha et al., 2024) | 96.4±2.9 | **98.9±1.1** | 92.4±4.2 | **95.5±3.0** |
| PCP-MAE (Zhang et al., 2024) | 94.0±4.8 | 97.8±1.9 | 89.3±5.2 | 93.9±3.0 |
| Point-CMAE (Ren et al., 2024) | 90.4±4.2 | 94.1±3.9 | 89.2±5.5 | 92.3±4.5 |
| Point-PQAE (Zhang et al., 2025) | 93.0±4.6 | 96.8±1.9 | 89.0±5.2 | 93.5±3.8 |
| **CSCon (Ours)** | **96.9±2.2** | 98.8±1.2 | 92.2±4.2 | **95.5±2.9** |
| *Improve (over Point-MAE)* | *+5.8* | *+7.1* | *+8.7* | *+5.8* |
| *with Self-Supervised Representation Learning (MLP − 3)* | | | | |
| Point-MAE (Pang et al., 2022) | 95.0±2.8 | 96.7±2.4 | 90.6±4.7 | 93.8±5.0 |
| ACT (Dong et al., 2022) | 95.9 ± 2.2 | 97.7 ± 1.8 | 92.4± 5.0 | 94.7 ± 3.9 |
| ReCon (Qi et al., 2023) | **97.4 ± 2.2** | 98.5 ± 1.4 | **93.6 ± 4.7** | 95.7 ± 2.7 |
| Point-FEMAE (Zha et al., 2024) | 96.4±2.9 | 98.9±1.1 | 92.4±4.2 | 95.5±3.0 |
| PCP-MAE (Zhang et al., 2024) | 96.7±2.7 | **99.1±0.9** | 92.9±4.2 | 95.7±2.7 |
| Point-CMAE (Ren et al., 2024) | 95.9±3.1 | 97.5±2.0 | 91.3±4.6 | 94.4±3.7 |
| Point-PQAE (Zhang et al., 2025) | 95.3±3.4 | 98.2±1.8 | 92.0±3.8 | 94.7±3.5 |
| **CSCon (Ours)** | **97.4±2.0** | **99.1±1.2** | 93.4±3.8 | **95.8±2.7** |
| *Improve (over Point-MAE)* | *+2.4* | *+2.4* | *+2.8* | *+2.0* |

LINEAR), where CSCon surpasses the strongest generative method, Point-MAE, by significant margins of +5.8%, +7.1%, +8.7%, and +5.8% across the different few-shot tasks.

**Object part segmentation.** We evaluate the representation learning capability of CSCon on ShapeNet-Part (Yi et al., 2016). ShapeNetPart, a widely used benchmark in 3D object segmentation, consists of 16,881 objects spanning 16 categories. During the experiments, we follow the same experimental settings and segmentation head as Point-MAE for a fair comparison. For evaluation, the mean Intersection over Union per category (Cls.mIoU, %) and the mean Intersection over Union over all instances (Inst.mIoU, %) are reported to provide a comprehensive assessment of model performance

Table 6: Part and semantic segmentation results on the ShapeNetPart and S3DIS Area 5 datasets: Mean intersection over union for all classes Cls.mIoU (%) and all instances Inst.mIoU (%) for Part Segmentation; Mean accuracy mAcc (%) and IoU mIoU (%) for Semantic Segmentation.

| Methods | Part Seg. | | Semantic Seg. | |
|---|---|---|---|---|
| | Cls.mIoU | Inst.mIoU | mAcc | mIoU |
| PointNet (Qi et al., 2017a) | 80.4 | 83.7 | 49.0 | 41.1 |
| PointNet++ (Qi et al., 2017b) | 81.9 | 85.1 | 67.1 | 53.5 |
| DGCNN (Wang et al., 2019) | 82.3 | 85.2 | - | - |
| PointMLP (Ma et al., 2022) | 84.6 | 86.1 | - | - |
| *with Self-Supervised Representation Learning (ALL)* | | | | |
| Transformer (Vaswani et al., 2017) | 83.4 | 84.7 | 68.6 | 60.0 |
| CrossPoint (Afham et al., 2022) | - | 85.5 | - | - |
| Point-BERT (Yu et al., 2022) | 84.1 | 85.6 | - | - |
| MaskPoint (Liu et al., 2022) | 84.4 | 86.0 | - | - |
| Point-MAE (Pang et al., 2022) | 84.2 | 86.1 | 69.9 | 60.8 |
| PointGPT (Chen et al., 2023) | 84.1 | 86.2 | - | - |
| Point-FEMAE (Zha et al., 2024) | 84.9 | 86.3 | - | - |
| PCP-MAE (Zhang et al., 2024) | 84.9 | 86.1 | 71.0 | 61.3 |
| Point-PQAE (Zhang et al., 2025) | 84.6 | 86.1 | 70.6 | 61.4 |
| **CSCon (Ours)** | 84.8 | 86.2 | 70.8 | 61.1 |
| *Improve (over Point-MAE)* | +0.6 | +0.1 | +0.9 | +0.3 |
| *Methods using cross-modal information and teacher models* | | | | |
| ACT (Dong et al., 2022) | 84.7 | 86.1 | 71.1 | 61.2 |
| ReCon (Qi et al., 2023) | 84.8 | 86.4 | - | - |
| *with Self-Supervised Representation Learning (ONLY − NEW)* | | | | |
| Point-MAE (Pang et al., 2022) | 83.7 | 85.1 | 66.2 | 52.5 |
| PCP-MAE (Zhang et al., 2024) | 83.5 | 85.3 | 66.1 | 54.1 |
| **CSCon (Ours)** | **83.9** | **85.5** | **68.3** | **56.0** |
| *Improve (over Point-MAE)* | +0.2 | +0.4 | +2.1 | +3.5 |

Table 7: Comparison of positive pairs.

| Methods | OBJ_BG | OBJ_ONLY | PB_T50_RS |
|---|---|---|---|
| Surrounding-Surrounding | 84.34 | 85.03 | 81.15 |
| Surrounding-Center (Ours) | 95.35 | 92.77 | 90.42 |

Table 8: Comparison of branch strategies.

| Methods | OBJ_BG | OBJ_ONLY | PB_T50_RS |
|---|---|---|---|
| Triplet-Branch | 92.43 | 91.22 | 88.76 |
| Dual-Branch (Ours) | 95.35 | 92.77 | 90.42 |

on 3D segmentation tasks. As shown in Table 6, under the ALL protocol, CSCon achieves comparable performance to existing methods and significantly outperforms Point-MAE (Pang et al., 2022). Under the ONLY-NEW protocol, CSCon achieves improvements of 0.2% and 0.4%, respectively.

**3D scene segmentation.** Semantic segmentation of large-scale 3D scenes is highly challenging, since it requires models not only to capture global semantic information but also to effectively model complex local detailed geometric structures. We conduct semantic segmentation experiments on the S3DIS dataset (Armeni et al., 2016), with detailed results presented in Table 6. All hyperparameters follow PCP-MAE. Under the ALL protocol, CSCon outperforms Point-MAE by 0.9% in mAcc and 0.3% in mIoU. Notably, under the ONLY-NEW protocol, our method achieves significant improvements over Point-MAE, with performance gains of 2.1% in mAcc and 3.5% in mIoU.

## 4.2 ABLATION STUDIES

**Different forms of positive pair.** One of the core contribution of our work is the construction of more variance positive pairs by separately masking center and surrounding points. This contrasts with previous contrastive methods, which overlook the importance of the center in a 3D patch. To validate the superiority of this design, we introduce a new ablation study: instead of our center-surrounding contrast, the model contrasts two sets of randomly masked surrounding points (*i.e.*, surrounding-surrounding). As shown in Table 7, this baseline performs far worse than our proposed CSCon, demonstrating the effectiveness of the positive pair constructing strategy of our CSCon.

**Ablation on alignment target.** CSCon is a contrastive learning method and is fundamentally different from masked reconstruction approaches. To verify this, we conducted a new experiment. First, we did not mask center and surrounding points, forming an input sequence $\mathbf{E}_s + \mathbf{E}_c$. Then, we

Table 9: Impact of parameter sharing strategies.    Table 10: Impact of varying contrastive objectives

| Type | OBJ_BG | OBJ_ONLY | PB_T50_RS |
|------|--------|----------|-----------|
| non-shared | 93.80 | 91.56 | 88.45 |
| shared | 95.35 | 92.77 | 90.42 |

| Granularity | OBJ_BG | OBJ_ONLY | PB_T50_RS |
|-------------|--------|----------|-----------|
| Inter-instance | 48.71 | 48.02 | 67.25 |
| Inner-instance | 95.35 | 92.77 | 90.42 |

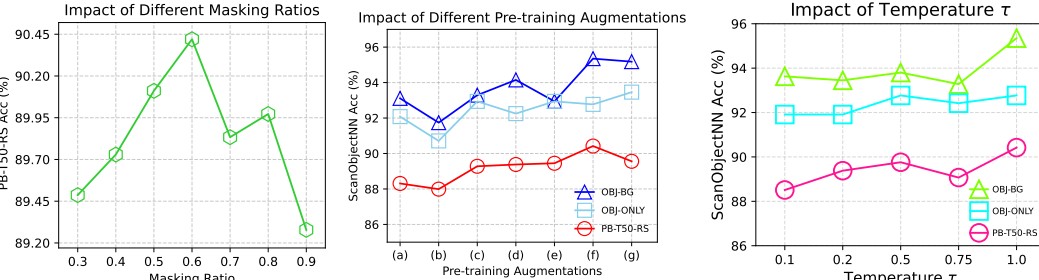

(a) Point-MAE      (b) Point-FEMAE      (c) PCP-MAE      (d) Ours

Figure 3: t-SNE (van der Maaten & Hinton, 2008) feature visualization on ScanObjectNN dataset, where the feature extracted by our CSCon is more concrete and discriminative than previous methods.

Figure 4: Performance evaluation of different masking ratios on the PB-T50-RS dataset.

Figure 5: Performance of different pre-training augmentations on ScanObjectNN dataset.

Figure 6: Impact of the temperature parameter $\tau$ on ScanObjectNN dataset.

separately masked center and surrounding points to create two sequences: $\mathbf{E}'_s + \mathbf{E}_c$ and $\mathbf{E}_s + \mathbf{E}'_c$. Theoretically, if CSCon were performing reconstruction, then $\mathbf{E}_s + \mathbf{E}_c$ should be able to guide both $E'_s + \mathbf{E}_c$ and $\mathbf{E}_s + \mathbf{E}'_c$. Therefore, we aligned the representation of $\mathbf{E}'_s + \mathbf{E}_c$ with that of $\mathbf{E}_c + \mathbf{E}_s$, and simultaneously aligned the representation of $\mathbf{E}_s + \mathbf{E}'_c$ with that of $\mathbf{E}_s + \mathbf{E}_c$. The experimental results are shown in Table 8. We found that the downstream task performance obtained through this alignment approach is far inferior to CSCon, which indirectly demonstrates that CSCon is performing contrastive learning rather than reconstruction.

**Parameter Sharing Strategies.** We also conduct a systematic analysis of parameter-sharing strategies for the encoder and projection head on the ScanObjectNN dataset. Specifically, we freeze all other hyperparameters. For a non-shared setting, we use two randomly initialized encoders and projectors, and the two modules are optimized via gradient descent separately. The results are shown in Table 9, our experimental results indicate that sharing parameters across the encoder and projection head significantly boosts model performance. We posit that the parameter-sharing mechanism effectively promotes consistent representations between the two branches in the feature space, reduces parameter redundancy, and enhances the model's generalization capability. Specifically, parameter sharing ensures that both branches maintain a unified representational format during feature extraction and projection. This facilitates the discrimination of positive and negative pairs in CL, thereby strengthening feature discriminability. Furthermore, parameter sharing mitigates the risk of overfitting, an effect that is particularly pronounced in scenarios with limited data or high task complexity. In contrast, while a non-shared architecture grants the model higher degrees of freedom, it may lead to the learning of inconsistent feature representations. This inconsistency weakens the constraints imposed by the contrastive loss, ultimately degrading overall model performance.

**Feature Visualization.** Figure 3 presents the t-SNE (van der Maaten & Hinton, 2008) visualization of feature manifolds, where t-SNE is configured to classify 15 categories with 2882 samples. The features are obtained after pre-training the model on ShapeNet and subsequently fine-tuning it on the ScanObjectNN PB-T50-RS dataset. Compared with other SOTA methods (Zha et al., 2024), CSCon yields the most discriminative features after fine-tuning on the downstream dataset. Contrastive

learning effectively pulls together features of samples from the same class while pushing apart those from different classes, significantly enhancing the discriminability of the feature space and resulting in clearer class boundaries in the t-SNE visualization compared to MAE. Furthermore, our proposed inner-sample patch-level contrastive loss further strengthens the representation of local features, enabling the model to better capture fine-grained local-detailed structural information and thereby improving both overall feature discriminability and downstream task performance.

**Varying Contrastive Objectives.** We compare the effects of contrastive losses at different granularities on classification performance. Specifically, we fix the other hyperparameter and change the training objectives. We design two objectives, i.e., inter-instance and inner-instance contrastive loss, where the inner-instance loss is shown in Eq. 7, and the inter-instance loss can be written as $\mathcal{L}_{Inter} = -\frac{1}{BM}[\sum_{i=1}^{BM} \log \frac{e^{sim(\mathbf{v}_c^i, \mathbf{v}_s^i)/\tau}}{\sum_{j=1}^{BM} e^{sim(\mathbf{v}_c^i, \mathbf{v}_s^j)/\tau}}]$. Where $B$ is the batch size, $M$ is the number of patches for contrasting in one sample, which means we take all the different patches in the same batch as negative pairs (both from the same sample and different samples). We find an interesting phenomenon, as shown in Table 10, where the inner-instance contrastive loss consistently outperforms the inter-instance counterpart across all metrics (inter-instance contrastive loss even does not learn any semantic information). We guess that's because the inter-instance contrastive loss potentially takes different patches both in the same point clouds and in different point clouds as the same negative pairs, while these two kinds of patches are semantically non-equivalent (patches from the same point clouds should be semantically closer than patches from different point clouds), leading to the collapse.

**Masking ratio.** In the pre-training stage, similar to PCP-MAE (Zhang et al., 2024), we conduct a set of experiments with different masking ratios to evaluate their impact on model performance. As shown in Figure 4, CSCon achieves the best performance when the masking ratio is set to 0.6.

**Pre-training Augmentation.** Effective pre-training augmentation strategies can significantly enhance the model's generalization ability and feature learning efficiency on complex 3D point clouds. To further investigate the impact of different augmentation strategies on the performance of CSCon, we conduct a series of ablation experiments with various combinations of data augmentations. Following the work Zhang et al. (2024), we conducted seven different data augmentation experiments (shown in Figure 5): (a) no data augmentation, (b) jitter only, (c) scale only, (d) rotation only, (e) scale & translate, (f) scale & translate combined with rotation, and (g) rotation combined with scale & translate. Specifically, we freeze all other hyperparameters and only change the augmentation. Our goal is to identify the optimal strategy that best enables CSCon to capture key features in high-dimensional space and improve downstream task performance. As shown in Figure 5, combining scale & translate with rotation achieves the best results, which is also consistent with PCP-MAE.

**Temperature parameter $\tau$.** To further analyze the impact of the temperature hyper-parameter $\tau$, we conduct an ablation study by sweeping over a range of values while keeping all other parameters fixed. The performance on the ScanObjectNN dataset, plotted in Fig. 6, indicates that the optimal result is achieved when $\tau = 1$.

## 5 CONCLUSION

In this paper, we propose a novel dual-branch center-surrounding contrastive learning framework, named CSCon, which partitions the patches into center and surrounding regions and treats these two incomplete parts as positive pairs, thereby forcing the network to learn rich geometric information. Furthermore, to enhance the learning of large-scale 3D local details, we design an inner-instance patch-level contrastive loss to learn high-level information. We conduct exhaustive experiments on standard self-supervised learning benchmarks, and the results demonstrate that our CSCon can extract both high-level and local-detailed information, achieving new SOTA results under both FULL, MLP evaluation protocols with much fewer parameters than previous SOTA.

**Limitations.** Currently, CSCon treats different patches within the same sample as negatives, scaling to scene-level point clouds with many patches may cause a sharp increase in the number of negatives.

**Future work.** The current pre-training relies solely on 3D point cloud data. A compelling direction is to incorporate multi-modal data (e.g., 2D images, text descriptions) during pre-training.

**LLM-Usage Statement.** The authors used a large language model for language polishing. The intellectual content, including ideas, methodology, and results, is solely the work of the authors.

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

# A    ADDITIONAL EXPERIMENTAL DETAILS

Table 11: Training details for pretraining and downstream fine-tuning.

| Config | ShapeNet | ScanObjectNN | ModelNet | ShapeNetPart | S3DIS |
|---|---|---|---|---|---|
| optimizer | AdamW | AdamW | AdamW | AdamW | AdamW |
| learning rate (FULL/ALL) | 5e-4 | 2e-4 | 1e-5 | 2e-4 | 2e-4 |
| learning rate (MLP-LINEAR/ONLY-NEW) | - | 6e-4/6e-4/2e-4 | 6e-4 | 5e-5 | 5e-5 |
| learning rate (MLP-3) | - | 2e-4 | 4e-4 | - | - |
| weight decay | 5e-2 | 5e-2 | 5e-2 | 5e-2 | 5e-2 |
| learning rate scheduler | cosine | cosine | cosine | cosine | cosine |
| training epochs | 300 | 300 | 300 | 300 | 60 |
| warmup epochs | 10 | 10 | 10 | 10 | 10 |
| batch size | 128 | 32 | 32 | 16 | 32 |
| drop path rate | 0.1 | 0.2 | 0.2 | 0.1 | 0.1 |
| number of points | 1024 | 2048 | 1024 | 2048 | 2048 |
| number of point patches | 64 | 128 | 64 | 128 | 128 |
| point patch size | 32 | 32 | 32 | 32 | 32 |
| augmentation | Scale&Trans+Rotation | Rotation | Scale&Trans | - | - |
| GPU device | RTX 4090/RTX 3090 | RTX 4090/RTX 3090 | RTX 4090/RTX 3090 | RTX 4090 | RTX 4090 |

**Pre-training dataset.** We pre-train CSCon on ShapeNet Chang et al. (2015), a large-scale 3D dataset comprising approximately 51,300 clean 3D models spanning 55 common object categories.

**Model Architecture.** We choose ViT-S as backbone, which consists of an encoder composed of 12 standard Transformer blocks (Vaswani et al., 2017) and a projection head implemented as a 2-layer MLP. Each Transformer block is equipped with 384-dimensional hidden units and 6 self-attention heads. The projection head outputs a representation with the same dimensionality as the encoder.

**Experiment Details.** During the pre-training stage, we follow established protocols for point cloud processing and model optimization. Specifically, input point clouds are first normalized via scaling and translation, followed by random rotations for data augmentation. We then uniformly sample 1024 points from each point cloud using FPS. To capture local geometric structures, the sampled points are further partitioned into 64 local blocks, each containing 32 points, using a combination of FPS and KNN. For model training, we pre-train CSCon for 300 epochs with a batch size of 128, employing the AdamW optimizer (Loshchilov & Hutter, 2017) with an initial learning rate of 0.0005 and a weight decay of 0.05. The learning rate is scheduled using cosine annealing (Loshchilov & Hutter, 2016), with a linear warm-up over the first 10 epochs to stabilize early training. All experiments are conducted on a single GPU, either an RTX 4090 (24GB) or RTX 3090 (24GB). Detailed experimental settings are provided in Table 11.

# B    TRANSFER PROTOCOL

**Transfer Protocols for Classification Tasks.**  Following the approach in (Dong et al., 2022), we use three variants of transfer learning protocols for classification tasks:

(a)  FULL: Fine-tuning pretrained models by updating all backbone and classification heads.

(b)  MLP-LINEAR: The classification head is a single-layer linear MLP, and we only update this head's parameters during fine-tuning, which evaluates the discriminative power of features.

(c)  MLP-3: The classification head is a three-layer non-linear MLP (which is the same as the one used in FULL), and we only update this head parameters during fine-tuning.

**Transfer Protocols for Part and Semantic Segmentation Tasks.**  We use two variants of transfer learning protocols for segmentation (also commonly used in previous methods (Zhang et al., 2024)):

(a)  ALL: Fine-tuning involves updating all the parameters used in downstream tasks of both the backbone and either the segmentation heads or the convolutional layers.

(b)  ONLY-NEW: Only the parameters of the segmentation head or the convolutional layers outside the encoder are updated, while the pretrained backbone parameters remain frozen.

**Transfer Protocol.** Following the approach in (Dong et al., 2022), we utilize three transfer learning protocols for the classification task: FULL, MLP-LINEAR and MLP-3. For the segmentation task,

we adopt two analogous protocols: ALL and ONLY-NEW. The specific details of these protocols are elaborated in the Appendix.

