# OpenReview forum: "Dual-Branch Center-Surrounding Contrast: Rethinking Contrastive Learning for 3D Point Clouds"
_ICLR.cc/2026/Conference — Submitted to ICLR 2026_

### Official Review · Reviewer_QAs4 · 2025-10-29

**Soundness:** 2
**Presentation:** 3
**Contribution:** 2
**Rating:** 4
**Confidence:** 5

**Summary:**

This paper proposes a novel self-supervised learning (SSL) framework for 3D point clouds, called CSCon (Dual-Branch Center-Surrounding Contrast). The authors argue that current 3D SSL methods are dominated by generative approaches (e.g., Masked Autoencoders), which fail to learn high-level discriminative features. To address this, CSCon introduces a dual-branch structure that separates center and surrounding patches of point clouds, coupled with an inner-instance patch-level contrastive loss. This design aims to improve both global discrimination and local geometric sensitivity. The method is extensively evaluated on multiple benchmarks, achieving state-of-the-art performance under several training protocols with fewer parameters.

**Strengths:**

+ Straightforward idea and design:  The center-surrounding dual-branch contrastive paradigm is intuitive. It effectively leverages the spatial structure of 3D point clouds for representation learning.

+ Clear methodology: The formulation of the inner-instance contrastive loss and ablation studies (mask ratio, data augmentation, parameter sharing) are comprehensive and well-explained.

+ Strong experimental results: The proposed method achieves consistent gains across multiple datasets (ScanObjectNN, ModelNet40, ShapeNetPart, S3DIS) and under various evaluation settings.

**Weaknesses:**

- While the method works well empirically, the paper lacks a strong theoretical justification for why the “center-surrounding” partition is optimal for contrastive learning. Could other partitioning schemes yield similar or better performance? For example, in [1], a perception-enlarged KNN strategy was proposed for patch generation, would it be beneficial for your method? Also, as a dual-branch SSL method, [1] should be cited.

[1] Chengzhi Wu, Qianliang Huang, Kun Jin, Julius Pfrommer, Jürgen Beyerer. A Cross Branch Fusion-Based Contrastive Learning Framework for Point Cloud Self-Supervised Learning. 3DV 2024.

- Self-supervised learning models are prone to falling into local optima or collapsing, particularly in contrastive learning methods. Various strategies have been proposed to mitigate this issue. For example, in pioneering works on image representation learning: (1) SimCLR and MoCo leveraged both positive and negative pairs; (2) MoCo and BYOL employed a momentum encoder; and (3) MoCo, SwAV, BYOL, and SimSiam adopted a stop-gradient operation on one branch. I am particularly surprised that the proposed method achieves successful training without employing any of these strategies, yet avoids collapse. Could you provide some insights into why this approach works? Additionally, please provide several top-tier conference or journal papers that report successful contrastive-based model training without relying on such specialized strategies.

- Ablation on loss temperature (τ)? Although the masking ratio was studied, other sensitive hyperparameters (e.g., τ in Eq.7) were fixed without justification.

- In the caption of Figure 2, shouldn’t it be that the left side displays the masked center points, and the right side shows the masked surrounding points?

- The definitions of the protocols are provided in the supplementary materials. Please note it in the main paper.

- Other minor writing issues. Some sections (especially the introduction and results) could be more concise, and grammar polishing could further improve readability.

**Questions:**

Please refer to the weaknesses, particularly the second point, as this will affect my decision regarding a possible score increase.

---

> ### Author Response · Authors · 2025-11-19
> **Rebuttal to Reviewer QAs4 (Part I)**
>
> Thank you for your careful review, thoughtful feedback, and constructive suggestions, which has been instrumental in helping us improve the quality of our paper.
>
>
> * Q1. Need theoretical justification and ablation studies on the optimal partitioning strategy.
>
> > Let us first consider a fundamental distinction between the 2D and 3D domains. In 2D, a patch consists of the representation derived from its internal pixels and the positional embeddings obtained from its patch index. In 3D, however, a patch includes the representation from all points within it, along with positional embeddings derived from the coordinates (xyz) of its center point.
>
> > Building on this, an interesting phenomenon was observed in PCP-MAE [1] (as shown in their Figure 1). Specifically, in the 2D domain, if all patches are masked in an MAE framework, reconstruction becomes impossible—because different images with identical pixels would share the same set of positional embeddings. In contrast, in 3D, even when all patches are masked, the entire point cloud can still be reconstructed. This is due to the fact that different point clouds have patches located at different positions, making the center point (xyz) of each patch unique. This indicates that in 3D, the center of each patch carries critical structural information.
>
> > Inspired by this, we proposed to explicitly separate the **center** and the **surrounding points** to construct positive pairs. By leveraging two distinct yet important attributes to form positive samples, we introduce stronger variance into the pairs—which is widely recognized as a key factor for successful contrastive learning. Our revised manuscript also includes the baseline of randomly masking the surroundings twice to form positive pairs. The results, presented in the table below, are substantially inferior to our center-surrounding approach. This confirms that the inherent variance between a center and its surroundings creates more meaningful positive pairs for contrastive learning.
>
> | Method | OBJ_BG | OBJ_ONLY | PB_T50_RS |
> | :--- | :--- | :--- | :--- |
> | Surrounding-Surrounding | 84.34 | 85.03 | 81.15 |
> | Surrounding-Center (CSCon) | **95.35** | **92.77** | **90.42** |
>
> > Additionally, we thank the reviewer for providing the highly relevant reference. We have now cited it in our revised manuscript. Following the suggested approach, we also conducted experiments using the Perception-Enlarged KNN strategy to construct positive samples. However, as shown in the table below, this method did not lead to improved performance in downstream tasks.
>
> | Method | OBJ_BG | OBJ_ONLY | PB_T50_RS |
> | :--- | :--- | :--- | :--- |
> | Perception-Enlarged KNN strategy+CSCon | 92.43 | 91.22 |  88.10 |
> | CSCon | **95.35** | **92.77** | **90.42** |
>
> [1] Zhang X, Zhang S, Yan J. Pcp-mae: Learning to predict centers for point masked autoencoders[J]. NeurIPS 2024.
>
> [2] Wu C, Huang Q, Jin K, et al. A cross branch fusion-based contrastive learning framework for point cloud self-supervised learning[J]. 3DV 2024.

---

> ### Author Response · Authors · 2025-11-19
> **Rebuttal to Reviewer QAs4 (Part II)**
>
> * Q2. Provide insights into the collapse-avoidance mechanism and related literature.
>
> > Thank you for this insightful question. As we know, a fundamental objective in contrastive learning is formulated by the InfoNCE loss [3], which aligns positive pairs while pushing apart negative samples. In [4], the loss is decomposed into two key components: **alignment** and **uniformity**. Alignment pulls positive samples closer, while uniformity encourages the representations of different samples to be distributed evenly on the unit hypersphere.
>
> > We would like to clarify that the underlying cause of model collapse lies in the alignment term (e.g., $||x - x^+||^2$), which can drive the model toward a constant solution—since outputting a constant value trivially minimizes this term. In CSCon, however, by treating other patches from the same instance as negative samples, we introduces a uniformity constraint within each instance. Clearly, a constant solution cannot satisfy this constrained objective, thereby effectively avoiding collapse.
>
> > Although patch-wise contrastive learning is rarely used in 2D and 3D domains, it has been widely adopted in graph representation learning. In graphs, an entire graph corresponds to a full point cloud (i.e., one instance), while nodes correspond to patches in point clouds or images. For example, in [5], the authors propose contrasting node representations from one view with the graph representation of another view. Furthermore, The work [6] and [7] explicitly use different nodes from the same graph as negative samples to avoid model collapse. Both of these approaches demonstrate that node/patch-wise contrastive learning (with intra-sample negative pairs) does not lead to instance-level collapse.
>
> [3] Oord A, Li Y, Vinyals O. Representation learning with contrastive predictive coding[J]. arXiv preprint arXiv:1807.03748, 2018.
>
> [4] Wang T, Isola P. Understanding contrastive representation learning through alignment and uniformity on the hypersphere[C]//ICML 2020.
>
> [5] Hassani K, Khasahmadi A H. Contrastive multi-view representation learning on graphs[C]//ICML 2020.
>
> [6] Zhu Y, Xu Y, Yu F, et al. Graph contrastive learning with adaptive augmentation[C]//WWW 2021.
>
> [7] Zhang S, Liu M, Yan J, et al. M-mix: Generating hard negatives via multi-sample mixing for contrastive learning[C]//SIGKDD 2022.

---

> ### Author Response · Authors · 2025-11-19
> **Rebuttal to Reviewer QAs4 (Part III)**
>
> * Q3. Ablation study on loss temperature $\tau$.
>
> > We initially set $\tau=1.0$ following established practices in 2D contrastive learning. To further address the reviewer's suggestion, we conducted an ablation study on $\tau$ (results shown in the table below and included in the Figure 6 of the revised version). The results demonstrate that $\tau=1.0$ yields robust performance, thereby empirically verifying the validity of our choice.
>
>
> | $\tau$ | OBJ_BG | OBJ_ONLY | PB_T50_RS |
> | :--- | :--- | :--- | :--- |
> | 0.1 | 93.63 | 91.91 | 88.51 |
> | 0.2 | 93.45 | 91.91 | 89.38 |
> | 0.5 | 93.8 | 92.77 | 89.76 |
> | 0.75 | 93.28 | 92.42 | 89.07 |
> | 1.0 | 95.35 | 92.77 | 90.42 |
>
>
>
> * Q4. More precise caption for Figure 2.
>
> > We thank the reviewer for their careful observation. The original caption for Figure 2 is indeed correct. We acknowledge that the 3D-to-2D projection in the original figure may have obscured the masking effect, leading to potential ambiguity. To address this, we have provided a new, clearer visualization in the revised manuscript (see Page 4, Figure 2) to accurately depict the masked centers and surrounding points.
>
>
>
> * Q5. Please include the definitions of the evaluation protocols in the main paper.
>
> > We have added "**Detailed evaluation protocols are provided in Appendix A.**"" in the revised version (see Page 5, Experiments section).
>
>
>
> * Q6. Improve writing and grammar.
>
> > Thank you for your suggestion, and we have correct the grammar and writing throughout the manuscript in the revised version (see the introcution and epxeriments sections).

---

> ### Author Response · Authors · 2025-11-23
> **Look forward to your further reply**
>
> Dear Reviewer QAs4
>
> Approaching the ending of the discussion phase, we wonder whether our response and additional results address your concerns and whether you have further questions about our revised version.

---

### Official Review · Reviewer_VRw5 · 2025-10-30

**Soundness:** 2
**Presentation:** 3
**Contribution:** 2
**Rating:** 4
**Confidence:** 4

**Summary:**

This paper presents a novel Dual-Branch Center-Surrounding Contrastive Learning (CSCon) method for 3D point cloud representation learning. The authors propose a framework where the point cloud is divided into center and surrounding parts, and these two parts are treated as positive pairs during contrastive learning. The approach includes patch-level contrastive loss to enhance the model's ability to capture both high-level discriminative features and fine-grained local details.

**Strengths:**

1.	Clear Expression: The paper is well-written, and the methodology is presented in a clear and logical manner.

2.	Easy to Follow: The paper is well-structured and easy to follow. The authors clearly explain the methodology, experimental setup, and results, ensuring that readers can easily understand the core concepts and contributions.

3.	Reasonable Complexity: The proposed method maintains reasonable complexity while achieving substantial improvements in performance.

4.	Solid Experimental Design: The experimental design is robust and well thought out. The paper provides clear validation of the method's performance across multiple datasets, and the results convincingly show the advantages of the CSCon approach over existing methods.

**Weaknesses:**

1.	Innovation Depth: The proposed innovation, where the encoded results remain consistent across different masking strategies for already partitioned patches, is relatively simple and straightforward. While it proves effective in improving performance, the novelty feels incremental when compared to the broader scope of 3D point cloud representation learning.

2.	Comparative Analysis: The authors predominantly compare their method with older works, which highlights the strengths of their approach. However, it would be valuable to include comparisons with more recent research (such as papers published in 2025). This would provide a more comprehensive understanding of where CSCon stands in relation to the latest advancements in the field and strengthen the paper's claim to state-of-the-art performance.

3.	Redundant Ablation Experiments: The first two ablation experiments appear somewhat redundant. These experiments could be moved to the supplementary materials to streamline the main content. Instead, it would be more valuable to include more essential ablation studies that directly address the key innovations of the method, as discussed in the Questions section.

**Questions:**

1.	Ablation Study:The current ablation studies focus on the impact of the full model. However, I would like to see a comparison where only one branch (either the center or surrounding branch) is kept. What would happen if we only use the first branch or the second branch in isolation?

2.	Mask Strategy and Powerful Baselines: The masking strategy is highlighted as the primary contribution of the paper. However, have you tested your method using more powerful baselines? This would help assess whether the improvements from the masking strategy are still significant when compared to stronger existing models.

3.	Broader Comparison with State-of-the-Art: While the comparisons made in the paper are insightful, I would recommend broader comparisons with more recent state-of-the-art models.

---

> ### Author Response · Authors · 2025-11-19
> **Rebuttal to Reviewer VRw5 (Part I)**
>
> We thank the reviewer for your positive feedback on the simplicity of our method and the solidity of our experiments, as well as for the constructive comments. Regarding the points raised, there appears to be some misunderstanding. Below, we provide a point-by-point response to address these concerns.
>
>
> * Q1. The proposed innovation is simple and incremental.
>
> > We emphasize that the contribution of CSCon is not the masking strategy. Rather, it lies in our deeper discovery of the critical importance of the center property in 3D point clouds—an aspect that has been almost entirely overlooked by previous contrastive learning methods. Recognizing the significance of the center, we **construct positive pairs by leveraging the center and its surrounding points**, which constitutes one of our key contributions.
>
> > To demonstrate the effectiveness of the proposed positive pairs, we designed a crucial ablation experiment: instead of masking the center in either branch, we kept the center unmasked in both and randomly masked surrounding points twice to generate positive pairs, thus performing "surrounding–surrounding" contrastive learning (note that this manner is actually the direct application of contrastive learning in point cloud).
>
> > The results of this experiment have been added to the revised version (in Table 8). For the reviewer's convenience, we include them again below. As shown in the table, this "surrounding–surrounding" design performs far worse than our proposed CSCon approach.
>
> | Method | OBJ_BG | OBJ_ONLY | PB_T50_RS |
> | :--- | :--- | :--- | :--- |
> | surrounding-surrounding | 84.34 | 85.03 | 81.15 |
> | CSCon | 95.35 | 92.77 | 90.42 |
>
> * Q2. Compare with more recent research.
>
> > In our revised version, we have included the results of several recently proposed methods from 2025 (as shown in Table 2 and Table 4). The results are also presented in the table below for clarity. It can be observed that our method can still achieve state-of-the-art performance while utilizing fewer parameters, demonstrating superior efficiency and effectiveness of our CSCon.
>
>
> | Methods | References| OBJ_BG | OBJ_ONLY | PB_T50_RS |
> | :--- | :--- |   :--- | :--- |:--- |
> | Point-FEMAE |AAAI 24  | 95.18 | 93.29 |90.22|
> | Point-PQAE | ICCV 25 |   95.0 | **93.6** | 89.6|
> | DAP-MAE | ICCV 25 |  95.18 | 93.45 |90.25|
> | Point-DMAE |CIKM 25 |   94.15 | 93.46 | 89.35|
> | UniPre3D |CVPR 25 |  92.60 | 92.08 | 87.93|
> | CSCon (Ours) |  | **95.35** | 92.77 | **90.42** |
>
> * Q3. Replace redundant ablations with more essential studies.
>
> > We thank the reviewer for their valuable suggestion. As ICLR allows the main text to be extended up to 10 pages during the rebuttal period, we have decided to retain this ablation study there. Meanwhile, in accordance with your suggestion, we have reordered all ablation studies in the revised manuscript.
>
> * Q4. Add ablation studies for each branch in isolation.
>
>
> > It is important to clarify that the dual-branch structure in our design does not consist of two independent feature modules. Rather, it is a necessary architecture within the contrastive learning paradigm to construct two augmented views to generate positive pairs. Consequently, operating a single branch in isolation is technically infeasible within our contrastive framework, as it would eliminate the positive pairs required to compute the contrastive loss.
>
> > However, we fully understand the reviewer’s core concern, which is to investigate the individual contributions of the center and surrounding masking strategies. To address this, we designed an alternative ablation study by evaluating these strategies within a generative framework. Specifically, we assessed the point cloud reconstruction performance when the model uses only center points information or only surrounding points information.
>
> > The results clearly demonstrate that both strategies perform excellently when used independently for reconstruction tasks. This strongly validates the effectiveness of our proposed masking strategies in capturing essential features.
>
> | Method | OBJ_BG | OBJ_ONLY | PB_T50_RS |
> | :--- | :--- | :--- | :--- |
> | Mask center only | 92.94 | 92.42 | 88.65 |
> | Mask surrounding only | 92.94 | 92.25 | 88.86 |

---

> ### Author Response · Authors · 2025-11-19
> **Rebuttal to Reviewer VRw5 (Part II)**
>
> * Q5. Compare with stronger baselines.
>
>
> > To further assess the generalizability of our method, we extended CSCon to PointMamba, an advanced architecture that has recently emerged. As a backbone based on State Space Models, PointMamba is highly representative of recent architectural developments.
>
> > It is important to note that, due to time and computational constraints during the rebuttal phase, we did not perform exhaustive hyperparameter finetuning for this new combination; instead, we adopted the basic configurations. Nonetheless, the experimental results (as shown in the table below) demonstrate that CSCon yields stable performance gains over the PointMamba baseline. This result strongly attests to the robustness of our method, indicating that it is not heavily reliant on specific parameter tuning.
>
>
> | Method | OBJ_BG | OBJ_ONLY | PB_T50_RS |
> | :--- | :--- | :--- | :--- |
> | PointMamba | 94.32 | 92.60 | 89.31 |
> | PointMamba+CScon | 95.15 | 93.27 | 90.30 |
>
>
> * Q6. Broader comparison with recent SOTA models.
>
> > Please refer to Table 2 and Table 4 in our revised version.

---

> ### Author Response · Authors · 2025-11-25
> **Look forward to your further reply**
>
> Dear Reviewer VRw5
>
> As we near the end of the discussion phase, we would like to confirm that our response and the additional results we provided have fully addressed your concerns. Please let us know if you have any further questions regarding the revised version.

---

### Official Review · Reviewer_zpkS · 2025-11-01

**Soundness:** 3
**Presentation:** 3
**Contribution:** 2
**Rating:** 4
**Confidence:** 4

**Summary:**

The paper introduces CSCon, a self-supervised 3D point cloud representation learning framework that contrasts center and surrounding patches within a dual-branch architecture. It abandons generative reconstruction (as in MAE-based models) and instead employs an inner-instance patch-level contrastive loss between masked center/surrounding features.

**Strengths:**

1, The paper correctly identifies the over-reliance of current 3D SSL on MAE-style reconstruction losses that learn low-level geometry but weak semantics. Addressing this via a contrastive objective that leverages 3D spatial structure is a meaningful idea.

2, Removing decoders and multi-view generation reduces computation and eases implementation.

3, This method achieves state-of-the-art performance.

**Weaknesses:**

1, Incremental conceptual novelty. The “center-surrounding” idea is intuitively similar to spatial partitioning already used in hybrid or region-aware methods (e.g., PointContrast’s local views, ReCon’s cross-patch contrast, Point-CMAE’s implicit local/global separation). The core innovation reduces largely to choosing intra-sample positives differently. Without a new theoretical insight or broader unification, the contribution is modest.

2, CSCon is seems like DetCo [1] in 3D,  local/global contrastive loss.

3, Unjustified reintroduction of patch-level contrastive learning. In 2D self-supervised literature, patch-level contrastive learning has been largely abandoned due to its semantic instability and inefficiency: local patches lack consistent meaning across samples, and strong augmentations make spatial correspondences unreliable, leading to noisy and contradictory gradients. Consequently, modern 2D frameworks (e.g., MAE, BEiT, DINOv2) have replaced patch-level InfoNCE with reconstruction or self-distillation losses that yield more stable local supervision. This paper claims that patch-level contrastive learning is more effective than MAE-style reconstruction in 3D, yet provides no theoretical explanation or empirical analysis clarifying why a contrastive objective—shown unstable in 2D—would suddenly become superior in the 3D domain. This weakens the conceptual credibility of the claimed improvement.


[1] DetCo: Unsupervised Contrastive Learning for Object Detection. ICCV 2021

**Questions:**

See weakness part

---

> ### Author Response · Authors · 2025-11-19
> **Rebuttal to Reviewer zpkS**
>
> We thank the reviewers for their thoughtful feedback and constructive suggestions. We believe there may have been some misunderstandings that we hope to clarify in our response.
>
>
> * Q1. Limited conceptual novelty.
>
> > CSCon differs fundamentally from existing contrastive learning paradigms. PointContrast generates different views by leveraging camera poses to construct positive pairs. ReCon applies contrastive loss to global information using cross-modal data. Point-CMAE constructs positive pairs by masking surrounding points of different patches. Most existing point cloud contrastive methods directly transfer 2D contrastive constraints to 3D point clouds, overlooking the inherent properties of 3D data. In contrast, **CSCon is built upon a core insight: constructing more challenging positive pairs by increasing their variance**—which is widely considered crucial in contrastive learning—to enhance learning effectiveness.
>
> > We observe that when a 3D point cloud is divided into patches, two key attributes emerge: the center point, which retains global information, and the surrounding points, which preserve local patch-level details. **CSCon explicitly leverages these two intrinsic attributes to perform contrastive learning.** To validate this design, we conducted an experiment where we directly applied Point-CMAE's contrastive framework to CSCon, that is, we randomly masked surrounding points in two different patches and applied a standard contrastive loss between them. As shown in the results table, this approach performs significantly worse than the CSCon paradigm. This empirically confirms the fundamental difference between CSCon and other contrastive learning methods at a conceptual and structural level.
>
> | Method | OBJ_BG | OBJ_ONLY | PB_T50_RS |
> | :--- | :--- | :--- | :--- |
> | Surrounding-Surrounding | 84.34 | 85.03 | 81.15 |
> | Surrounding-Center (CSCon) | **95.35** | **92.77** | **90.42** |
>
>
> * Q2. Clarify novelty over DetCo [1].
>
> > Although both CSCon and DetCo employ patch-level contrast and share certain similarities, their approaches to constructing positive pairs are fundamentally different. DetCo learns dense features by cropping local 2D views, whereas the core objective of CSCon is to mine challenging positive pairs in 3D—specifically, by leveraging the two inherent attributes of **center** and **surrounding**.
> In the 2D domain, when an image with the same pixel content is divided into patches, these patches share the same patch index, i.e., a label that carries very limited semantic meaning. Therefore, it is not feasible to perform contrastive learning between a patch index (positional embeddings) and its corresponding patch representation. In contrast, in 3D, the patch index corresponds to the center point of a point cloud patch, which inherently contains global structural information. This key distinction makes CSCon's global-local contrast fundamentally different from DetCo's global-local contrast.
>
>
>
> * Q3. Needs justification for reintroducing 3D patch-level contrastive learning.
>
> > We thank the reviewer for this insightful observation. The analysis regarding the instability of patch-level contrastive learning in 2D is indeed accurate. We fully agree that the failure of 2D patch-level contrastive learning can be largely attributed to its reliance on random cropping as a core data augmentation strategy.
>
> > As you pointed out, random cropping results in inconsistent correspondence of patches across different views, i.e., a patch in one view may correspond ambiguously or to multiple patches in another randomly cropped view. This unreliable spatial alignment introduces instability during training, which explains why the 2D community has gradually shifted towards reconstruction or self-distillation methods.
>
> > However, unlike in the 2D domain, **CSCon does not rely on cropping to construct positive pairs**. Instead, we take a complete point cloud as input and decompose it into patches. Our positive pairs are formed from two inherent attributes of the same patch, i.e., the **center point** and the **surrounding points**. This design ensures strict one-to-one correspondence between every positive pair. Consequently, CSCon avoids the semantic instability issues that plague 2D contrastive learning due to unreliable spatial correspondence.

---

> ### Author Response · Authors · 2025-11-25
> **Look forward to your further reply**
>
> Dear Reviewer zpkS:
>
> We are hopeful that our detailed response and the additional data have satisfactorily resolved the points raised. As we approach the final stage of this discussion, please don't hesitate to share any remaining questions you may have about the revisions.

---

### Official Review · Reviewer_dY7K · 2025-11-01

**Soundness:** 3
**Presentation:** 2
**Contribution:** 2
**Rating:** 6
**Confidence:** 5

**Summary:**

In this paper, the authors propose a dual-branch center-surrounding contrastive learning method to pre-train 3D point cloud models. They point out that existing mask auto-encoding methods performs poorly on linear probing and that contrastive learning mechanism in 3D remains undeveloped. The proposed CSCon framework first split input point cloud into centers and corresponding point patches. Then a dual branch framework encode and mask centers and patches separately. Transformer blockes are leveraged to predict the mask regions. Finally, contrastive loss is performed between two branches to realize a fine-grained patch-level contrastive learning. The authors conduct extensive experiments on various benchmarks and achieves promising results.

**Strengths:**

1. The proposed methods combines contrastive learning and MAE-style pre-training, realizing fine-grained patch-level reasoning. The method is technically sound and conceptually interesting.

2. The experiment results show relatively strong performance on various benchmarks.

**Weaknesses:**

1. The method is only limited to object-level datasets.  A major difference between contrastive learning and MAE-style pre-training is that contrastive learning can be better applied to more complex scene-level scenarios. Since the authors categorize their method to contrastive-based method, the missing experiments of pre-training directly on scene-level datasets like ScanNet would largely undermine the strength of the paper.

2. It would be better if the author could analyze more thoroughly into the difference between these feature groups with ablation studies:
(1) Es & Vs & Ec'+Es & Es'+Ec (2) Ec & Vc & Es'+Ec & Ec'+Es. From the method, Vc and Vs seem like reconstruction of Es'+Ec and Ec' + Es. However, since no point cloud is explicitly reconstructed, the readers would have no idea about what Vc and Vs actually encoded. More thorough analysis and experiments into this issue will reveal more in-depth insight of the proposed method.

**Questions:**

1. I'm confused by Figure 2, the visualization on the ShapeNet set. The figure shows masking ratio of 40%, 60% and 80%. However, the masked region in the figure seems much smaller than expected. For example, the random mask 40% sample only masks a tiny part of the bottom of the plane, while the random mask 80% sample only masks approximately 40% of the sample.

2. What is the resource and time consumption of this method? It would be better if efficiency could also be mentioned besides params used.

3. Missing related work comparison:

[1] Gao, Ziqi, Qiufu Li, and Linlin Shen. "DAP-MAE: Domain-Adaptive Point Cloud Masked Autoencoder for Effective Cross-Domain Learning." Proceedings of the IEEE/CVF International Conference on Computer Vision. 2025.

[2] Lin, Xuanyu, et al. "PointLAMA: Latent Attention meets Mamba for Efficient Point Cloud Pretraining." arXiv preprint arXiv:2507.17296 (2025).

[3] Wang, Ziyi, et al. "UniPre3D: Unified Pre-training of 3D Point Cloud Models with Cross-Modal Gaussian Splatting." Proceedings of the Computer Vision and Pattern Recognition Conference. 2025.

[4] Cheng, Haozhe, et al. "PointFM: Point Cloud Understanding by Flow Matching." IEEE Robotics and Automation Letters (2025).

---

> ### Author Response · Authors · 2025-11-19
> **Rebuttal to Reviewer dY7K**
>
> Thank you for your insightful questions and valuable suggestions. Here are our point-by-point responses.
>
>
> * Q1. More experiments on ScanNet dataset.
>
> > We have initiated pre-training experiments on the ScanNet dataset. However, since existing baseline methods do not include experiments on this dataset, we lack directly comparable implementations and well-tuned hyper-parameters (such as the learning rate). Given the time constraints of the rebuttal period, we may not be able to report final results at this stage. Nevertheless, we are committed to completing the experiments as soon as possible and will provide updated results in the final version, which will include the full pre-training outcomes.
>
> * Q2. More ablation studies and analysis on the feature groups are needed.
>
> > Actually, $V_c$ and $V_s$ are not reconstructing $E_s' + E_c$ and $E_c' + E_s$. The core idea of CSCon originates from contrastive learning. Consider the 2D image domain: when two random crop views are taken from an image and a contrastive loss is applied, are the two crops reconstructing the original image? Similarly, CSCon divides patches into two subsets, i.e., center and surrounding, is somewhat analogous to image cropping in 2D, except here we are “cropping” two attributes of 3D patches.
>
> > To verify this, we conducted a new experiment. First, we did not mask center and surrounding points, forming an input sequence $E_s + E_c$. Then, we separately masked center and surrounding points to create two sequences: $E_s' + E_c$ and $E_s + E_c'$. Theoretically, if CSCon were performing reconstruction, then $E_s + E_c$ should be able to guide both $E_s' + E_c$ and $E_s + E_c'$. Therefore, we aligned the representation of $E_s' + E_c$ with that of $E_c + E_s$, and simultaneously aligned the representation of $E_s + E_c'$ with that of $E_s + E_c$. The experimental results are shown below. We found that the downstream task performance obtained through this alignment approach is far inferior to CSCon, which indirectly demonstrates that CSCon is performing contrastive learning rather than reconstruction.
>
> | Method | OBJ_BG | OBJ_ONLY | PB_T50_RS |
> | :--- | :--- | :--- | :--- |
> | $(E_s' + E_c)$->$(E_s + E_c)$ and $(E_s + E_c')$->$(E_s + E_c)$ | 92.43 | 91.22 | 88.76 |
> | CSCon | 95.35 | 92.77 | 90.42 |
>
> > We sincerely appreciate your constructive question, which gave us the opportunity to clarify our method. We have already incorporated these results and discussion into the revised version of our paper (in Page 8, Table 8).
>
>
> * Q3. Figure 2's masking visualization seems incorrect.
>
> > Thanks for your careful reviews. We have updated the 3D visualizations in the revised version of our paper to clearly reflect the actual extent of information removal in the point cloud at different masking ratios.
>
>
> * Q4. Report computational efficiency.
>
> > Beyond the model parameter counts already compared in the main body of the paper, we further provide crucial evidence regarding our method's efficiency, including the actual time consumption during the pre-training phase and the downstream task fine-tuning phase. All efficiency comparison experiments were conducted on a single RTX 4090 GPU, and the specific results are presented in the table below. The experimental data clearly indicate that our CSCon method not only maintains the lowest number of model parameters but also significantly surpasses the computational efficiency of most existing methods during both the pre-training and fine-tuning stages. This further underscores the practical utility and superiority of our approach, particularly in resource-constrained environments.
>
> | Methods | Params (M) | Time (s/epoch) | OBJ-BG Time (s/epoch) | OBJ-ONLY Time (s/epoch) | PB-T50-RS Time (s/epoch) |
> | :--- | :--- | :--- | :--- | :--- | :--- |
> | Point-MAE | 29.0 | 107 | 12 | 12 | 53 |
> | Point-FEMAE | 41.5 | 157 | 16 | 16 | 72 |
> | CSCon | **22.1** | **90** | **6** | **6** | **22** |
>
>
> * Q5. Add comparison with the listed related works.
>
> > We thank the reviewer for bringing these highly relevant concurrent works to our attention. We have incorporated them into our references and added comparisons with these methods in Table 2 and Table 4 of our revised version.

---

### Author Response · Authors · 2025-11-20
**General Response to All reviewers and ACs.**

Dear Area Chair and Reviewers,

We sincerely appreciate your thorough feedback and constructive suggestions. Overall, the reviewers have recognized our work's well-founded motivation (dY7K, zpkS, QAs4), clear presentation (VRw5, QAs4), and the effectiveness of our method (dY7K, zpkS, VRw5, QAs4), which is supported by comprehensive experimental results (dY7K, zpkS, VRw5, QAs4). We note that the main concerns raised by the reviewers primarily focus on the theoretical foundation of the contrastive paradigm (zpkS, QAs4), conceptual novelty (zpkS, VRw5), and the need for additional ablation studies and comparative analyses (dY7K, VRw5, QAs4).

In response to these valuable comments, we have made point-by-point revisions in the updated manuscript (with all changes highlighted in blue for easy reference). Here is a summary of the key updates:

* Incorporate new ablation studies on surrounding-surrounding contrast.
* Add ablation studies on using PointMamba backbone.
* Provide the computational cost and time during both pre-training and fine-tuning stages.
* Provide a theoretical explanation on the necessity of introducing uniformity constraints by treating other patches from the same sample as negative samples.
* Included additional comparative results with the latest methods published in 2025.
* Conduct ablation studies on the hyperparameter sensitivity of the temperature coefficient $\tau$.

---

### Meta-Review · Area_Chair_a4bw · 2025-12-29

**Summary:**

The paper proposes CSCon, a dual-branch self-supervised learning framework for 3D point clouds. It replaces the generative MAE objective with a proposed patch-level contrastive loss. Based on the comments and responses, the current version has unresolved concerns regarding novelty and theoretical grounding, making it unsuitable for acceptance so far. For instance, the method partitioning patches into centre and surroundings is viewed as a minor variation of existing spatial partitioning strategies, such as Point-CMAE. The reviewers found the conceptual innovation limited, noting that the method largely reapplies established contrastive principles without a fundamental breakthrough. Besides, there is a lack of robust theoretical explanation for why this architecture avoids model collapse without standard stabilisers like momentum encoders or large negative batches. Reviewer noted that patch-level contrastive learning was largely abandoned in 2D due to instability, and the authors' responses for its success here remains empirically driven.

**Reviewer Concerns:**

The authors successfully provided efficiency analysis and corrected visualization issues, for Reviewer `dY7K` and Reviewer `QAs4`. They also included missing recent baselines and related works requested by Reviewer `VRw5` and Reviewer `QAs4`. Additionally, the requested ablation on the temperature hyperparameter was provided. The concern regarding technical novelty remains unresolved, with Reviewer `zpkS` and Reviewer `VRw5` viewing the center-surrounding partition as a minor, incremental variation of existing strategies. Furthermore, Reviewer `QAs4`’s and Reviewer `zpkS`’s concerns about the lack of theoretical justification for why the model avoids collapse without standard stabilisers were not sufficiently answered, as the rebuttal relied on empirical observations. Reviewer `dY7K`'s request for full scene-level (ScanNet) pre-training results also remains constraint.

**Reviewer Scores:**

Reviewer `dY7K` (init score 6) is likely to lower the rating, since the authors provided the requested efficiency analysis, the continued absence of the critical scene-level (ScanNet) pre-training experiments remains a significant gap. The other reviewers might likely maintain their position, as the rebuttal clarifying the differences with DetCo did not fundamentally resolve the concern that reintroducing patch-level contrastive learning lacks theoretical justification given its known instability in the 2D domain.

---

### Decision · Program_Chairs · 2026-01-26

Reject